# Charge Characteristics of Dielectric Particle Swarm Involving Comprehensive Electrostatic Information

**DOI:** 10.3390/mi14122151

**Published:** 2023-11-24

**Authors:** Yue Feng, Xingfeng Shen, Ruiguo Wang, Zilong Zhou, Zhaoxu Yang, Yanhui Han, Ying Xiong

**Affiliations:** 1School of Mechatronical Engineering, Beijing Institute of Technology, Beijing 100081, China; 3220235218@bit.edu.cn (X.S.); 3220220133@bit.edu.cn (R.W.); 3120205118@bit.edu.cn (Z.Z.); 3120235418@bit.edu.cn (Z.Y.); 2Electrostatic Business Department, Beijing Orient Institute of Measurement and Test, Beijing 100094, China; hui920718@163.com; 3Laboratoire Catalyse et Spectrochimie, École Nationale Supérieure d’Ingénieurs de Caen, Université de Caen, CNRS, 14050 Caen, France; ying.xiong@ensicaen.fr

**Keywords:** particle swarm, discrete element method, charge characteristics, electrostatic

## Abstract

The triboelectrification effect caused by dynamic contact between particles is an issue for explosions caused by electrostatic discharging (ESD) in the triboelectric nanogenerators (TENGs) for powering the flexible and wearable sensors. The electrostatic strength of dielectric particles (surface charge density, surface potential, electric field, etc.) is essential to evaluate the level of ESD risk. Those differential electrostatic characteristics concerned with unhomogenized swarmed particles cannot be offered via in-current employed-joint COMSOL 6.1 simulation, in which the discrete charged dielectric particles are mistakenly regarded as continuous ones. In this paper, the hybrid discrete element method (EDEM tool) associated with programming in COMSOL Multiphysics 6.1 with MATLAB R2023a was employed to obtain the electrostatic information of the triboelectric dielectric particle swarm. We revealed that the high-accuracy strengths of electric potential and electric field inside particle warm are crucial to evaluating ESD risk. The calculated electrostatic characteristics differ from the grid method and continuous method in the surface potential and electric field. This EDEM-based simulation method is significant for microcosmic understanding and the assessment of the ESD risk in TENGs.

## 1. Introduction

Rotary sliding mode triboelectric nanogenerators (TENGs) based on particle motion provide high efficiency and a high output strategy for harvesting low-frequency mechanical energies [1,2,3,4,5,6]. The harvested energy can be used for powering the flexible and wearable sensors. When the particles rub and rotate in the polytetrafluoroethylene tube, the contact surface can produce a high charge density. Only the triboelectrification between the particles and the wall was considered, and the triboelectrification between the particles was ignored in the past. The electrostatic discharging (ESD) generated by a high charge density may break down the electric field of the dielectric medium [7,8,9,10]. The ESD of the dielectric particles may lead to spark discharging [11], brush discharging [12], and broad bulk discharging inside the TENGs [13], as well as trigger an explosion while TENGs are placed in the chemical industry. Moreover, the ESD between particles causes intergranular ESD between the particles and the wall, which greatly reduces the harvested energy. Therefore, it is necessary to understand the generation of electrostatic charges and ESD.

Many experiments have been conducted to investigate the electrostatic characteristics [14,15,16,17], and the cone discharging from highly insulating bulked polyethylene granules was solved using radio frequency signals and a sudden drop in Ref. [18]. The typical values in a single discharge range from 20 to 50 μC. In addition, the equivalent discharging energy of 10 mJ was obtained by diverting the cone discharging to a spark gap [13]. This previous work provided an effective method for studying dielectric discharging. T. Suzuki et al. described the mechanism of the electrification of particles via numerical approaches [19] and experimentally investigated the electrostatic discharging [20,21]. An electrostatic ionizer with a modeling test device was developed to measure the charge-neutralizing current. Four-typed discharging (brush, linear, broad, and dot) were observed during continuous powder loading, which differs in terms of generation time and duration [22]. The results showed that the charge-to-mass ratio increased as the powder feeding rate decreased, and the risk of ESD was dramatically reduced as the net charge decreased.

The finite element methods (FEM) are essential to accessing the electrostatic performance of dielectric particles [23,24,25,26]. It is a difficult to identify the accurate results of electric potential and electric field through FEM simulation. The simulation of saturated particle accumulation showed that ESD risk existed even at a shallow stacking height [24]. However, the simulation method is based on a two-dimensional plane. The distribution of electric potential with particles was studied in Ref. [25]. Liang et al. [26] simulated the electric field distribution in a bench-scale silo containing charged polyethylene particles and found the maximum values located in the positions of the heap surface, the wall, and the heap bottom. The simulation cannot show the comprehensive information (positions, radius, surface charge densities, etc.) of the particle swarm. The discrete element methods (DEM) are effective in revealing detailed information about the infinite number of particles. However, the generation of electric potential and electric field among individual particles cannot be obtained in discrete element tools and by failing to judge the level of ESD. Liang et al. employed the grid method to import the particle information obtained from EDEM into the COMSOL tool [27]. The unhomogenized electrostatic property inside the particle swarm was tentatively simulated by dividing the regions and counting the amount of charge in different regions in this method. However, it is hard to provide the data regarding air gap, particle size, and surface charge density, resulting in the inaccurate estimation of the electric field strength.

In this paper, the coupling simulation of DEM and FEM was effectively set up to describe the electrostatic characteristics of a dielectric particle swarm including comprehensive information (positions, radius, surface charge densities, etc.). The results of electric potential and electric field can be obtained more accurately by characterizing the comprehensive information of the particle swarm. The remaining parts of this paper are arranged as follows: Section 2 describes the simulation principle and boundary settings of the DEM and FEM in the triboelectrification effect of particles. Section 3 analyzes the comparison of three simulation methods (i.e., the continuous method, the grid method, and the particle method). Section 4 presents the conclusions of this work.

## 2. Simulation Conditions

### 2.1. Simulation Methods

The continuous method is the most commonly used way to study the electric potentials and electric fields of particles. The discrete particles are treated as closely stacked and continuous balls, which adopt the factor of space charge density. However, its equivalent model has several issues. Firstly, the space charge density in the continuous method is fully obtained by dividing the surface charge density of a dielectric particle by its volume. Furthermore, the air domains between the particles are deemed to be particle domains, leading to overlarge simulated charges. In addition, the lack of air domains greatly reduces the electric potential of contact surfaces between particles and sidewalls.

The grid method is proposed to solve the uneven particle distribution by counting the total charge of a specific region and dividing that charge by the volume to determine the space charge density [27]. The single-layer meshing is expanded to the multi-layer one, and their irregular regions are divided. This method can split the grid into tiny ones to solve the problem of an uneven charge distribution. The dielectric constant of the air is much lower than that of the insulating particles, which means ESD in the air medium occurs much earlier. However, the grid method cannot predict the ESD of the air medium due to the lack of air domains.

Our developed particle method can provide the particle position, radius, and surface charge density of the particles. This comprehensive information is essential for the prediction of the specific ESD type (brush, linear, broad, and dot). The flow chart of our proposed particle method-based simulation is shown in Figure 1. Firstly, the above necessary data obtained from EDEM are imported into COMSOL 6.1 for geometric modeling, condition setting, and calculation. Since it is a difficult task to model a large number of particles in COMSOL 6.1 alone, this paper writes instructions based on COMSOL Multiphysics 6.1 with MATLAB R2023a to complete the simulation. The key codes are shown in Appendix A. Also, the usage of MATLAB R2023a Environmental to control COMSOL Multiphysics 6.1 can automatedly model particles in COMSOL 6.1 and greatly improve the simulation efficiency and accuracy. This simulation strategy fully considers air domains between particles, which enables more consistency with the actual conditions. In addition, the method of mesh division in different areas is controlled by codes in MATLAB R2023a that reduce the number of mesh and reduce the calculated time. Figure 2 shows the particle method depicting the densities of particles, air domain, and surface charge. In Figure 2a, the red circles represent particles, while the yellow circles represent air domains. Figure 2b shows a schematic diagram of the surface charge conditions.

### 2.2. Simulation Setup

The cylindrical stainless-steel container (radius 120 mm, height 120 mm) is modeled as a particle container, as shown in Figure 3. The spherical air domain of 400 mm in radius is considered at infinity as having zero potential. Table 1 provides the necessary property parameters of the silica particles. The range of particle radius is from 1.7 to 2.3 mm, and the surface charge density of the particles is distributed from 0.5 to 1.5 μC/m^2^.

Figure 4 shows the distribution results of the 50,000 particles in the container. Figure 4a show that the height of the deposited particles is 67.7 mm and the radius of the container is 120 mm, and Figure 4b shows that the diameter of the particles ranges from 3.4 to 4.6 mm. The particle charges in Figure 4c ranges from 1.8 × 10^−11^ C to 1.0 × 10^−10^ C.
(1)Q=4πrρ

The triboelectric charges of particles can be obtained using the discrete element tool EDEM. Only the accurate information (position, radius, and charge) of the particle can ensure the accuracy of the potential and electric field obtained in the finite element software COMSOL 6.1.

## 3. Results and Discussion

### 3.1. Particles of the Same Radius and Surface Charge Density

Here, the condition of the same particle radius and surface charge density are used for the simulation of the past research. The continuous method is based on the tight packing of particles, and the stacked particles are assumed to be the same dielectric. The radius and surface charge density of the particle are set to be 2 mm and 1 μC/m^2^, respectively. The two-layer boundary of the spherical infinite element field is grounded.

The simulated electric potential and electric field in the Y-Z plane corresponding to the three simulation methods are shown in Figure 5. The “Threshold” on the scale bars (3 × 10^6^) of the simulation data represents the threshold values for dielectric breakdown. Exceeding these thresholds can result in reduced charge due to dielectric breakdown. The increment trends of electric potential and electric field are similar for the continuous method, grid method, and particle method. The regions of large electric potential simulated via those methods are located on the surface where particles come into contact with air domains. The electric potential decreases from the container center to the sidewall in the X-Y plane and decreases from the contact surfaces to the container bottom in the Z direction. The large electric field strength tends to concentrate on the surfaces where the particles come into contact with the containers. And the largest one is found in the bottom center of the container.

The electric potential given via the continuous method with 50,000 particles in Figure 5a is greatly larger than those calculated from the other two methods in Figure 5b,c. The values of generated charge and electric potential in Figure 5a are nearly three times those in Figure 5b,c. The space charge density is obtained by dividing the surface charge density of a single particle by its particle volume in the continuous method. However, there are a large number of air gaps among the stacked particles, resulting in air domains exhibiting unexpected charging effects in the continuous method. The total amount of carried charges is calculated to be 7.145 × 10^−6^ C, which is significantly larger than those of 2.514 × 10^−6^ C given via the other methods. The continuous method widely used in previous studies is adapted to predict the locations of the highest electric potential and the strongest electric field with short time consumption. However, the continuous method cannot provide the threshold of ESD of air domains. As a result, its accuracy is still an issue in terms of evaluating the ESD level. The electric field strength in Figure 5c is about three times larger than that of Figure 5b because the particle method intentionally takes air domains into account owing to different dielectric constants of the air domain and the dielectric particles.

Figure 6 shows the maximum values of particles in the Y-Z plane. The maximum electric potential given by the grid and particle methods is always equivalent in quantity in the number range of 10,000~50,000. It is proved that both the grid method and the particle method are more accurate in the calculation of electric potential. As the number of particles increases, the thickness of particle accumulation increases. The simulation method, considering the characteristics of the air domain itself, produces a more significant size effect, that is, the electric field value is significantly different to that of the other two methods.

However, it is difficult for particles to carry the same surface charge density in real life, so the research under this condition has no great practical application value.

### 3.2. Particles of the Same Radius and Different Surface Charge Densities

The complete consistency of particle swam much be satisfied using the continuous method. Here, only the grid and particle methods are compared with respect to the electric potential and electric field with different surface charge densities and the same radius. The new cross-sections, including the X-Y plane (bottom) and X-Y plane (top), are selected to better reveal the ESD locations and ESD forms.

The following results are derived from our proposed particle method. Figure 7 and Figure 8 show the simulation results of the electric potential and electric field with 50,000 particles (higher stacked particles). It is shown in Figure 7a and Figure 8a that the level of ESD near the container bottom is relatively high with respect to the calculated intensity of the electric field. The ESD occurs not only in the regions between particle sidewalls but also in the regions between particles. With the aid of the individual modeling of air domains, the electric field strengths of air domains enable nonuniform distribution, which determines the ESD types. Moreover, the electric field strength of the 50,000 stacked particle does not exceed 3 × 10^6^ V/m, and the ESD does not occur; in contrast, much higher stacked particles of 50,000 cause the electric field strength to grow up to 6 × 10^6^ V/m, which is sufficient to trigger ESD. It is clearer to explain the advantages of the particle method in view of the X-Y plane shown in Figure 7b and Figure 8b. The particle method involving air domains gives a much higher electric potential on the bottom surface of the container compared to the continuous and grid methods without the air domains. The electric potential *U* at the bottom center of the container is 200~250 V in Figure 7b; the electric potential *U* at the bottom center of the container is 600~650 V in Figure 8b. The electric potential *U* of the container is equal to 0 V at the location of the grounded container. The electric field strength *E* calculated according to *E* = *U*/*Z* is similar to the simulation result of electric field strength. The difference between the electric potential values can mainly be attributed to the following two causes: the dielectric constants of the air and particles are not the same, and the contacts forming between particles and between the particles and container are the point contacts in the particle method, while the grid method adopts the surface contact. In the previous simulation study, when a single particle was in contact with the wall, the electric field strength of the surface contact (square particle) was significantly smaller than that of the point contact (spherical particle). The electric field strength is more concentrated under the point contact. Moreover, there should be no charge inside the particle as an insulating dielectric, and the charge is only bound to the particle surface. There should be no electric field strength inside the particle, and the setting of space charge density makes the field strength inside the particle, which is inconsistent with reality. The reason that the other two methods cannot accurately calculate the electric field is that they recognize the field charge distribution as the space charge density, but this is inconsistent with reality. Moreover, the different surface charge densities make the distribution of the electric field unhomogenized. In Figure 8b, the regions with the darkest color of the electric field strength are not only concentrated closest to the center but also in the outer two or three cells. There are also several lighter red dots in a pile of darker red dots in Figure 7b because the total amount of charge carried by these particles is significantly smaller than those of the surrounding particles.

The electric potential distribution in the X-Y plane (top) was found to be the same by comparing Figure 7c and Figure 8c. The closer to the center, the greater the electric potential. The electric potential is 0 V at the edge of the container wall. The two simulation methods are also closer in terms of numerical results. The electric field strength near the container wall of both methods is greater. The distribution of different charge densities makes the simulation result of electric field strength decrease completely symmetrically at the bottom of the container. The electric field results obtained are shown to be quite different both in the value and distribution of the grid and particle methods by comparing Figure 7c and Figure 8c. The electric field strength of the particle method is three times that of the grid method, which has reached the critical value of 3 × 10^6^ V/m of air discharge. In addition, the regions with high electric field strength are close to the wall in the grid method. There is also a large electric field strength region between the two particles in the central position of the particle method. The simulation results are in better agreement with the actual experiments [9,28]. Cone discharging is the phenomenon of discharges occurring along the surface of highly charged bulked polymeric granules, as shown in Figure 9a; tree discharging is shaped like branches, as shown in Figure 9b; and large brush discharging is a feather-like shape, as shown in Figure 9c. It is mentioned in this study that the cone discharge can easily occur where the particles contact the container wall, and the large brush discharge and tree discharge can easily occur near the central surface of the particles. The electric field strength results obtained via the particle method are in good agreement with cone discharge, large brush discharge, and tree discharge. In the trend results of the grid method, only the cone discharge between the particle and the wall of the container can be observed, and the electric field strength in the center is much smaller than that at the wall. Therefore, only the cone discharge can be obtained. The particle method has the advantage of more discharge phenomena being observed on the contact surface of the particle and air domain compared to the grid method.

The maximum electric potential increases with the increase in the particle number, while the maximum electric field strength is similar at 40,000 and 50,000 particles. When the material accumulation height in the container reaches a certain value, the electric field strength does not significantly increase, which is also consistent with the previous simulation results [23]. While the electric field strength of the particles increases with the increase in height in the grid method, the simulation results of the grid method incorrectly show that the maximum electric strength does achieve 3 × 10^6^ V/m, and ESD hazards do not occur. It is easier for the ESD hazard to occur when the surface charge density of particles is found to be closer by comparing the electric field strength with the same surface charge density and different surface charge densities.

### 3.3. Particles of Different Radius and Surface Charge Densities

The influence of particle radius on electric potential and electric field has rarely been studied in previous studies due to the fact that previous simulation models could not characterize the radii of individual particles. The grid method is omitted in this section because it cannot study the effect of particle size on the electric field. This section used the particle method to reveal how different radii of particles affect the electric potential and electric field with the working conditions (same radius and surface charge density, same radius and different surface charge density, different radius and surface charge density).

The simulation results are shown as the X-Y plane (middle) and X-Y plane (top) in Figure 10. The maximum electric field strengths of particles in the X-Y plane are from 4 × 10^6^~5 × 10^6^ V/m and from 3.5 × 10^6^~4 × 10^6^ V/m, respectively, for the middle and top positions. The unhomogenized distribution of particle radius increases the electric field strength through the simulation results shown in Figure 11c,d. The particle with the largest charge is 1.0 × 10^−10^ C, while the particle with the smallest charge is 1.8 × 10^−11^ C, as shown in Figure 4c. When the two particles happen to be close to each other, a large electric potential difference is generated, and further calculations give a larger electric field strength. The larger electric field strength indicates that ESD is more likely to occur in containers. In addition, the stacked particles are more densely packed in Figure 11d than in Figure 11c. Figure 11c,d have brighter regions, which indicates that when discharge occurs on the surface of particles in the container, particle groups with different surface charge densities are likely to produce large-scale and large-area discharging phenomena.

Table 2 shows the maximum electric field strength of the four sections under three different working conditions. It can be seen from the table that particles of the same radius and the same surface charge density are the most dangerous for containers. The most dangerous is at the X-Y plane (middle), where the electric field strength is 141% of the same radius and there is a different charge density. This indicates that discharge hazards are most likely to occur when the surface charge density of particles in the container is close. The occurrence of discharge hazards can be effectively avoided by making the surface charge density of particles in the container different.

### 3.4. Simulation Methods Discussion

The three simulation methods used in this paper are summarized. Their applicability conditions and advantages are pointed out as follows:The continuous method can be used in the condition of the same radius and surface charge density. This method can be used to obtain the trend of electric potential and electric field strength for the cone ESD occurring between particles and container walls. This method has the simplest modeling and the lowest requirement for grid division among the three methods, so it only needs a very short simulation time to complete, but the accuracy is unsatisfactory.The grid method can be used in conditions of the same radius and different surface charge densities. The grid method can obtain the accurate electric potential value and the uneven electric potential. The grid method needs a shorter time compared to the particle method, and it can also be used first in large-scale particle simulation to determine the location of the maximum potential and the maximum electric field strength distribution, but it cannot be employed to simulate particles with different radii.The particle method based on the combination of DEM with FEM can provide comprehensive information about particles. This method solves the problem stemming from the fact that the influence of particle radius on the electric field cannot be studied via the previous simulation. The air domain and point contact in actual working conditions obtain a more accurate electric field strength, which is useful to determine the occurrences of large brush charging and tree charging.

The present simulation results suggest the following measures to better avoid ESD hazards: 1. Reduce the packed thickness of particles in the container. 2. Increase the humidity of the air in the container. 3. Use particle materials that are less prone to charging.

## 4. Conclusions

In this paper, the triboelectric ESD caused by dynamic contact between particles is an issue in TENGs powering flexible and wearable sensors. An electrostatic simulation strategy based on the hybrid discrete element method associated with programming in COMSOL 6.1 was developed for unhomogenized swarmed particles. The essential information generated from EDEM was successfully imported into COMSOL 6.1, in which the accurate model reconstruction was realized with comprehensive information. Taking the silica particles deposition inside a container as an example, the electric field strength of container particles in the particle method differs from those in the continuous method and the grid method. The maximum electric field strength of 8.26 MV/m given by the particle method exceeds the breakdown threshold of air (3 MV/m), while the other two methods are still mistakenly considered to be safe. The particle method provides both the air domain and particle domain, while only the particle domain without the air domain is calculated using the other two methods. Also, the point contact in the particle method is more accurate compared to the surface contact from the other two methods. In addition, the particle method can determine the occurrences of cone discharging, brush discharging, and tree discharging, while the other two methods only obtain the occurrence of cone discharging. Of course, only the particle method can characterize the difference in particle radius, and the simulation results of the particle method are consistent with the actual process state.

## Figures and Tables

**Figure 1 micromachines-14-02151-f001:**
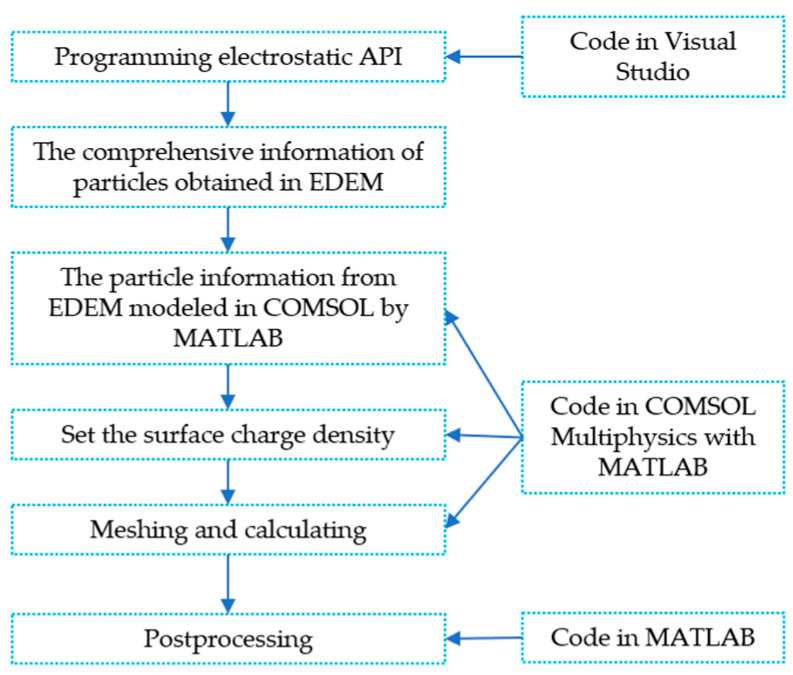
The flow chart and some key codes of particle method-based simulation.

**Figure 2 micromachines-14-02151-f002:**
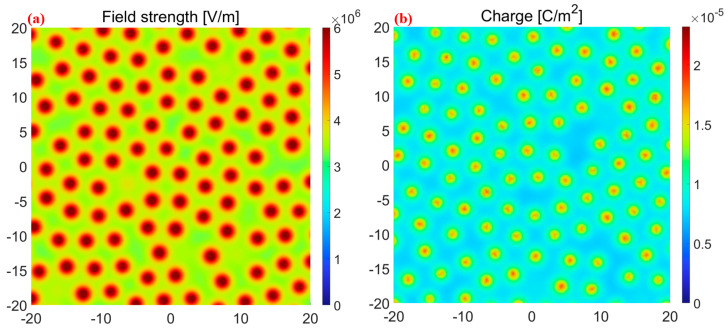
Each particle interacts with (**a**) the air medium and (**b**) the surface charge.

**Figure 3 micromachines-14-02151-f003:**
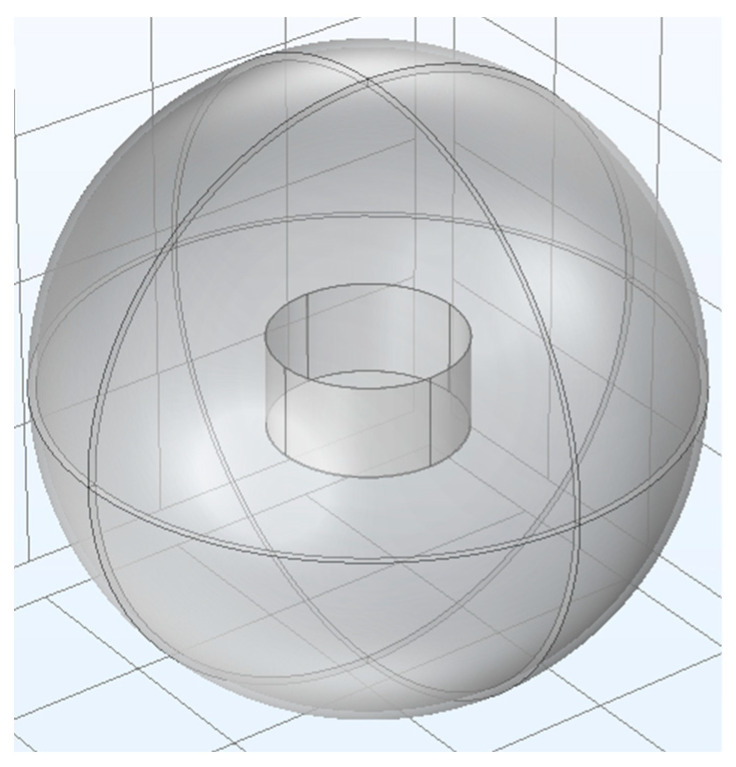
Geometric model of container and air domains.

**Figure 4 micromachines-14-02151-f004:**
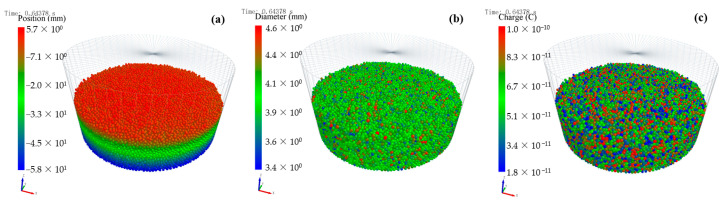
The comprehensive information regarding particle swarm in EDEM: (**a**) position, (**b**) radius, and (**c**) surface charge density.

**Figure 5 micromachines-14-02151-f005:**
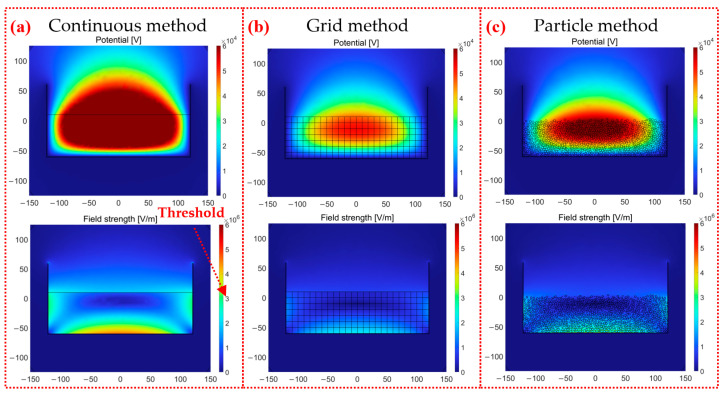
The electric potential and electric field strength in the Y-Z plane with 50,000 particles: (**a**) the continuous method, (**b**) the grid method, and (**c**) the particle method.

**Figure 6 micromachines-14-02151-f006:**
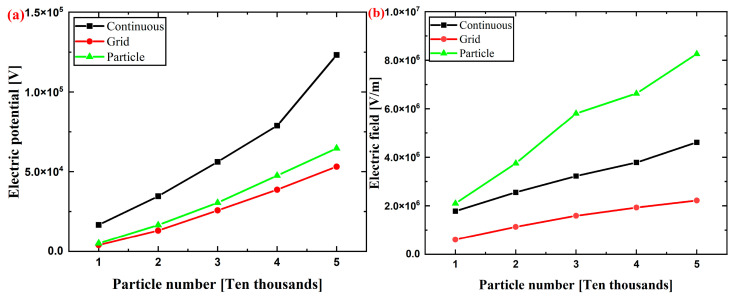
The maximum values in the YZ plane of the continuous, grid, and particle methods in the number range of 10,000~50,000: (**a**) electric potential and (**b**) electric field.

**Figure 7 micromachines-14-02151-f007:**
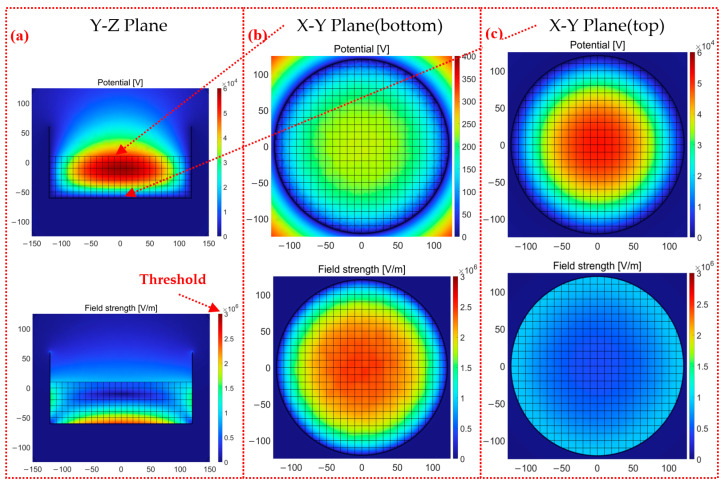
The electric potential and electric field strength of the grid method with 50,000 particles: (**a**) Y-Z plane, (**b**) X-Y plane (bottom), and (**c**) X-Y plane (top).

**Figure 8 micromachines-14-02151-f008:**
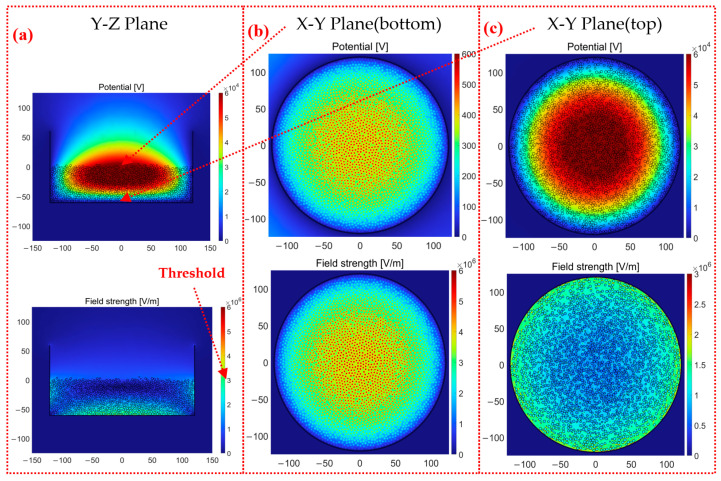
The electric potential and electric field strength of the particle method with 50,000 particles: (**a**) Y-Z plane, (**b**) X-Y plane (bottom), and (**c**) X-Y plane (top).

**Figure 9 micromachines-14-02151-f009:**
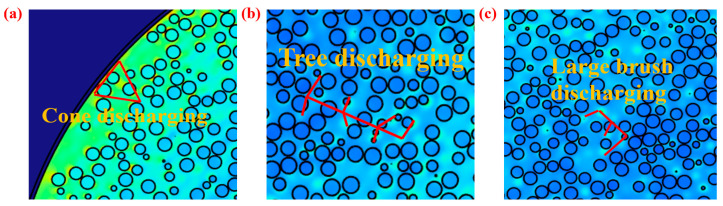
The ESD types: (**a**) cone discharging, (**b**) tree discharging, and (**c**) large brush discharging.

**Figure 10 micromachines-14-02151-f010:**
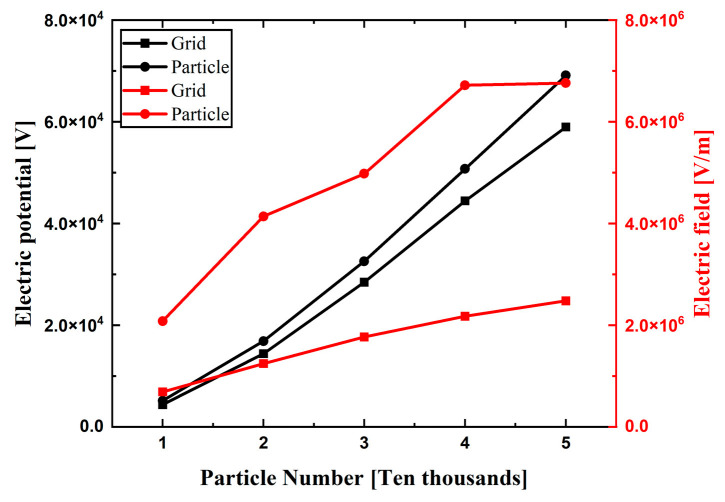
The maximum values of the electric potential and electric field strength of the grid method and particle method are in the number range of 10,000~50,000.

**Figure 11 micromachines-14-02151-f011:**
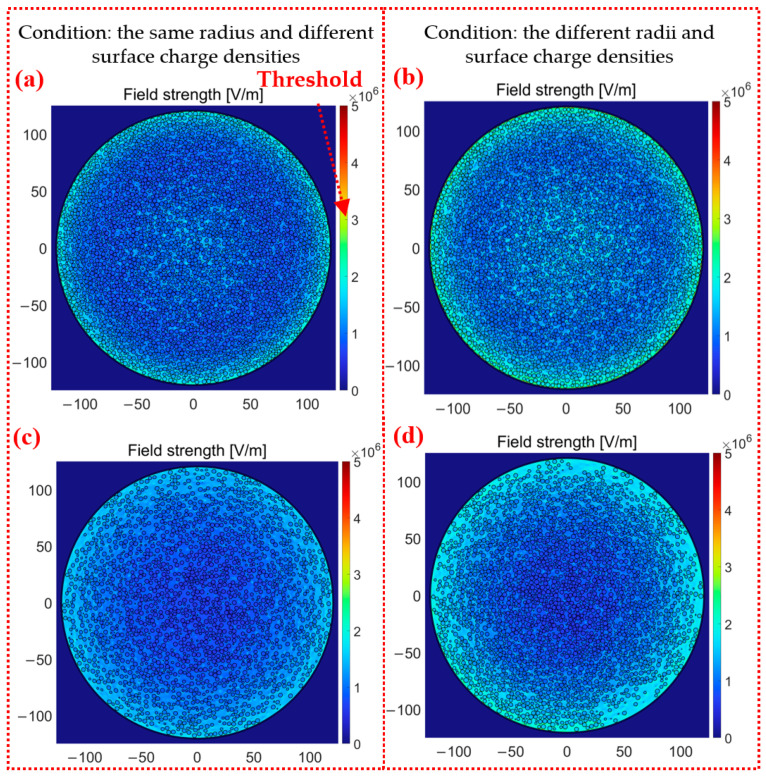
The electric field strength of the particle method with 50,000 particles: (**a**,**b**) X-Y plane (middle) and (**c**,**d**) X-Y plane (top).

**Table 1 micromachines-14-02151-t001:** Particle properties used in DEM and COMSOL 6.1 simulations.

Particle Parameter	Value
Shape	Sphere
Radius (mm)	1.7~2.3
Density (kg/m^3^)	650
Poisson ration	0.3
Young’s modulus (MPa)	25
Surface charge density (μC/m^2^)	0.5~1.5
Relative dielectric constant	1.88

**Table 2 micromachines-14-02151-t002:** The simulation results of the electric fields.

	Y-Z Plane	X-Y Plane (Bottom)	X-Y Plane (Middle)	X-Y Plane (Top)
Same radius and surface charge density (MV/m)	8.26	10.43	5.74	4.22
Same radius and different surface charge densities (MV/m)	6.76	7.68	4.05	3.26
Different radius and surface charge densities (MV/m)	7.90	9.24	5.05	3.99

## Data Availability

Data are contained within the article.

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
