# Peer review of "Charge Characteristics of Dielectric Particle Swarm Involving Comprehensive Electrostatic Information"

_micromachines, 2023, doi:10.3390/mi14122151_

Round 1

Reviewer 1 Report

Comments and Suggestions for Authors

The often-used discrete element method (EDM) is significant to evaluate the level of ESD risk if the information of electrostatic charges and electrostatic strength cane be revealed. This work developed an improved EDM strategy associated with programming, aiming to obtain comprehensive charge characteristics of swarmed dielectric particle. The clear advantages of presented continuous method is persuasive after comparison with the grid method and the particle method. Some comments should be considered to improve the quality of this manuscript.

1. The contribution of programing code in COMSOL is quite unique to obtain comprehensive charge characteristics, but the author couldn’t show the details of MATLAB programing. It is necessary to add some key codes in Figure 1.

2. In the 3.2 part, the author emphasized this simulation method can determine the ESD level and then the types, such as core discharge, large brush discharge, and tree discharge (Line 243). It is interesting if the ESD types can be demonstrated with the figures. Please explain the ESD types, concepts and their differences.  Also, adding more information in Figures (enlarged) to show how to distinguish and determine ESD types.

3. Please add some suggestions that how to prevent the ESD according to present simulation results, since the particle numbers (thickness of particle swarm ) would influence the electric field strength.

Comments on the Quality of English Language

Minor editing of English language required

Author Response

The authors are very grateful for the reviewers' critical comments and thoughtful suggestions. Based on the comments and suggestions, the authors have made careful modifications to the original manuscript. The following are the responses to the reviewers' comments.

Comment 1 response: Thanks very much to the reviewer's suggestions, some key codes of MATLAB programming have been added as shown in Fig. 1. The entire modeling process was completed by repeatedly executing the MATLAB code for a single ball. The key code will help reviewers and readers better understand the modeling process.

Comment 2 response: Thanks very much for the reviewer's advice. Cone discharging is the phenomenon of discharges occurring along the surface of highly charged bulked polymeric granules. Tree discharging resembles branches, while large brush discharging takes on a feather-like shape. The enlarged figures that demonstrate how to distinguish and determine ESD types are shown in Figure 9.

Comment 3 response: Thank you very much for the reviewer's comments. The following three methods can be used to prevent ESD: reducing the packed thickness of particles, increasing the air humidity, and using particle materials that are not easily charged.

Reviewer 2 Report

Comments and Suggestions for Authors

This paper presents a new scenario that can adjust the overestimation of charge quantity in TENG simulations utilizing particle-shaped charged materials. In the field of TENGs, calculations of potential and electric field regarding surface charge are often overestimated due to fundamental errors, diminishing reliability. Thus, research that identifies and provides solutions for such overestimations is highly significant. Overall, the flow and interpretation of the calculations are logically organized, but certain elements might cause confusion among readers. So, I have a few comments on the expression methods.

1.       The concept is hard to understand. An additional diagram or figure explaining how each particle interacts with air and surface charge conditions would significantly aid understanding. This could be a crucial factor for other researchers when deciding whether to reference this paper's concept.

2.       In all figures, the scale bars for potential and electric field are different. Simply looking at the colors can be misleading, as it appears to be the opposite of what is described in the manuscript. Authors should use the same scales for results with the same physical meaning (e.g., potential with potential, electric field with electric field). Adjustments can be easily made using the “range” list in COMSOL's result “plot” window. If the differences are too big, using a logarithmic scale could also be a good approach. Particularly for electric fields, it might be beneficial to display up to the threshold of air breakdown or note the breakdown point on the scale bar.

3.       In Figures 6 and 7, it's unclear what constitutes the top and bottom. Indications on the Y-Z plane would be helpful.

4.       Adding the threshold values of dielectric breakdown to the scale bars of the simulation data, visually indicating that exceeding these thresholds could lead to reduced charge due to dielectric breakdown, would greatly assist non-experts in understanding.

Author Response

The authors are very grateful for the reviewers' critical comments and thoughtful suggestions. Based on the comments and suggestions, the authors have made careful modifications to the original manuscript. The following are the responses to the reviewers' comments.

Comments 1 response: Thank you very much for the reviewer's comments. The author includes Figure 2 in the paper. The particles are represented by the red part, while the air domains are represented by the yellow part in Fig. 2(a). The surface charge conditions are depicted in Fig. 2(b). 

Comments 2 response: Thanks to the reviewer's advice, the author has redrawn the color bars for the electric potential and electric field. This was done to ensure that the upper limit of the color bars remains consistent within the same picture. The author inevitably includes the original color bars in order to accurately depict the detailed information obtained from the simulation in certain figures.

Comments 3 and 4 response: Thanks very much for the reviewer's suggestions. The X-Y Plane (bottom) and X-Y Plane (top) are indicated in the Y-Z Plane by red arrows. We have added the threshold values of dielectric breakdown to the scale bars in the simulation data, and we have also provided an explanation of the threshold in the paper.
